# Long-Term Consequences of COVID-19: A 1-Year Analysis

**DOI:** 10.3390/jcm12072673

**Published:** 2023-04-03

**Authors:** Laurence Bamps, Jean-Philippe Armenti, Mirela Bojan, Bruno Grandbastien, Christophe von Garnier, Renaud Du Pasquier, Florian Desgranges, Matthaios Papadimitriou-Olivgeris, Lorenzo Alberio, Martin Preisig, Jurg Schwitter, Benoit Guery

**Affiliations:** 1Service of Infectious Diseases, Lausanne University Hospital and University of Lausanne, 1011 Lausanne, Switzerland; 2Service of Internal Medicine and Infectious Diseases, Cliniques Universitaires Saint-Luc, Avenue Hippocrate 10, 1200 Brussels, Belgium; 3Anesthesiology, Hopital Marie Lannelongue, 133 Av. de la Résistance, 92350 Le Plessis-Robinson, France; 4Division of Pneumology, Lausanne University Hospital and University of Lausanne, 1011 Lausanne, Switzerland; 5Service of Neurology, Lausanne University Hospital and University of Lausanne, 1011 Lausanne, Switzerland; 6Service of Haematology and Haematology Central Laboratory, Lausanne University Hospital and University of Lausanne, 1011 Lausanne, Switzerland; 7Service of Psychiatry, Lausanne University Hospital and University of Lausanne, 1011 Lausanne, Switzerland; 8Service of Cardiology, Lausanne University Hospital and University of Lausanne,1011 Lausanne, Switzerland; 9Director CMR Center, University Hospital Lausanne (Centre Hospitalier Universitaire Vaudois (CHUV)), 1011 Lausanne, Switzerland; 10Faculty of Biology & Medicine, Lausanne University, 1011 Lausanne, Switzerland

**Keywords:** long COVID-19, SARS-CoV-2

## Abstract

Long-lasting symptoms after SARS-CoV-2 infection have been described many times in the literature and are referred to as Long COVID. In this prospective, longitudinal, monocentric, observational study, we collected the health complaints of 474 patients (252 ambulatory and 222 hospitalized) at Lausanne University Hospital 1 year after COVID-19 diagnosis. Using a self-reported health survey, we explored cardiopulmonary, vascular, neurological, and psychological complaints. Our results show that age, Charlson comorbidity index, and smoking habits were associated with hospital admission. Regarding the vascular system, we found that having had thromboembolism before SARS-CoV-2 infection was significantly associated with a higher risk of recurrence of thromboembolism at 1 year. In the neurologic evaluation, the most frequent symptom was fatigue, which was observed in 87.5% of patients, followed by “feeling slowed down”, headache, and smell disturbance in 71.5%, 68.5%, and 60.7% of cases, respectively. Finally, our cohort subjects scored higher overall in the STAI, CESD, Maastricht, and PSQI scores (which measure anxiety, depression, fatigue, and sleep, respectively) than the healthy population. Using cluster analysis, we identified two phenotypes of patients prone to developing Long COVID. At baseline, CCS score, prior chronic disease, stroke, and atrial fibrillation were associated with Long COVID. During COVID infection, mechanical ventilation and five neurological complaints were also associated with Long COVID. In conclusion, this study confirms the wide range of symptoms developed after COVID with the involvement of all the major systems. Early identification of risk factors associated with the development of Long COVID could improve patient follow-up; nevertheless, the low specificity of these factors remains a challenge to building a systematic approach.

## 1. Introduction

Severe Acute Respiratory Syndrome Coronavirus 2 (SARS-CoV-2) was first reported in Wuhan, China, in December 2019 [1]. During 2020, this pathogen spread across the world, creating a worldwide pandemic that is ongoing at this time, with more than 757 million cases and almost 7 million deaths having been reported at the time of writing [2]. Coronavirus disease 2019 (COVID-19) is associated with a wide spectrum of symptoms, ranging from asymptomatic to severe pneumonia, requiring invasive ventilation, and death [3,4,5]. It has also been shown that COVID-19 is a multifaceted disease with different forms and presentations, including cardiovascular [6], neurological [7], dermatological, and gastrointestinal symptoms [8] as well as other extrapulmonary manifestations [9].

In the early stage of the pandemic, the main concern was the initial severity and immediate related mortality that was mostly a consequence of respiratory failure [10,11]. Interestingly, with an increasing number of patients recovering from COVID-19, its long-term consequences are being rapidly described. Long-lasting symptoms have already been described for several other viral diseases, such as influenza [12] and recent coronavirus epidemics (MERS- and SARS-CoV-1) [13]. A large proportion of COVID-19 survivors still suffer from many symptoms weeks or months after recovery. Davis et al. analyzed an online survey of 3762 participants, and the most frequent symptoms were fatigue, post-exertional malaise, and cognitive dysfunction [14]. Several other studies also confirmed persistent neuro-cognitive and neurological symptoms after COVID-19 [15,16]. Long-term respiratory sequelae are also commonly described [17]. A prospective longitudinal cohort study of patients admitted for severe COVID-19 in Wuhan followed 135 individuals from 3 to 12 months after discharge [18]. At 12 months, a subgroup of patients still presented a decreased diffusing capacity of the lungs for carbon monoxide (DLCO) as well as radiological changes, underlining the need for long-term follow-up of these patients. Another cohort study, also performed in China, enrolled 1733 patients with a median follow-up of 186 days [19]. Again, the most common symptoms were fatigue or muscle weakness, but the authors also demonstrated psychiatric consequences including anxiety and depression associated with pulmonary alterations (impairment of DLCO). Globally, neurological, pulmonary, psychiatric, and respiratory consequences have been reported as a consequence of COVID-19 during the long-term follow-up of patients in several studies [20,21,22,23]. Recently, a 1-year outcome trial in hospital survivors of COVID-19 in Wuhan studied 1276 patients [24]. The authors evaluated and reported health-related quality of life, physical examination, 6 min walk test, and laboratory test results at 12 months. The proportion of patients with at least one sequelae symptom decreased from 68% at 6 months to 49% at 12 months, but more patients had anxiety and depression at 12 months. Compared to the control population, the health status of COVID-19 survivors was significantly lower. In another study, Frontera et al. showed that life stressors such as financial insecurity, food insecurity, death of a loved one, or disability were strong predictors of worse long-term cognitive, neuropsychiatric, and functional outcomes 12 months after COVID-19 hospitalization [25].

An increasing body of literature supports the idea that there is a chronic clinical entity that includes a wide array of symptoms that can be defined as Long COVID, which has more recently collectively been referred to as post-acute sequelae of SARS-CoV-2 infection (PASC). However, there is currently no specific and accepted definition of this entity. In 2020, the National Institute for Health and Care Excellence (NICE) in the UK proposed to define Long COVID as “signs and symptoms that develop during/after the COVID-19 infection persisting for more than 4 weeks and could not be explained by any other diagnosis” [26]. The symptoms are not clearly described or classified and involve a mixture of cardiovascular, respiratory, and neurologic domains. Baig et al. proposed a different term, chronic COVID syndrome, the diagnosis of which is based on organ threats [27]. In this model, the authors proposed an organ stage-based classification after 3 weeks. Interestingly, only one article has proposed diagnostic criteria for Long COVID; clinical and duration criteria were described for asymptomatic and symptomatic patients to categorize cases as confirmed, probable, possible, or doubtful [28]. Even in this article, symptom was a generic term and not clearly defined. Fernadez de las Penas et al. suggested the terms long post-COVID symptoms (from 12 to 24 weeks) and persistent post-COVID symptoms (more than 24 weeks) [29]. These authors suggested a temporal relationship between COVID-19 and the symptoms that appear after infection. In a second model, these authors included the dynamics of the disease, including exacerbation, delayed-onset, and persistent symptoms [30]. This last approach included, for the first time, the fluctuation or relapsing nature of post-COVID-19 symptoms. In a systematic review analyzing persistent symptoms reported in 39 studies up to March 2021, Michelen et al. [31] suggested that Long COVID is a syndrome characterized mostly by fatigue, weakness, malaise, breathlessness, and concentration impairment, among other less frequent symptoms.

In this study, we report the symptoms and health complaints of COVID-19 survivors that were admitted with COVID-19 at a Swiss university hospital. In order to address the multifaceted nature of the disease, we estimated the prevalence of symptoms in four organ systems (respiratory, cardio-vascular, neurologic, and psychiatric) via a self-reported survey at 12 months compared to the initial symptoms presented on diagnosis of acute COVID-19. After assessing the current symptoms and complaints about patient health, we linked these to the initial acute COVID-19 presentation. We finally performed clustering analysis to characterize the profile of Long COVID patients in order to provide a clinical-based definition of this entity.

## 2. Materials and Methods

### 2.1. Study Setting

This study originated from Lausanne University Hospital (CHUV), a 1500-bed tertiary university hospital in Lausanne, Switzerland.

### 2.2. Study Design and Participants

This prospective, longitudinal, monocentric, observational study included all adult patients with a confirmed SARS-CoV-2 infection from 1 March 2020 to 31 December 2020 (first wave and beginning of second wave of COVID-19, thus before the detection of variants of concern in Switzerland, https://www.covid19.admin.ch/fr/epidemiologic/virus-variants, accessed on 9 March 2023). Patients fulfilling the following criteria were considered eligible and were sent a written informed consent form and questionnaire: age ≥ 18 years old, tested for COVID-19 infection documented with a positive reverse transcriptase–polymerase chain reaction (RT-PCR) assay for SARS-CoV-2 in a respiratory tract sample, and alive at 12 months. Those who signed the informed consent were included.

The survey consisted of four different questionnaires covering specific areas of potential post-COVID-19 persistent symptoms. Respiratory and cardiac symptoms were assessed using the New York Heart Association (NYHA) classification and the Canadian Cardiovascular Society (CCS) angina score [32]. Vascular status was assessed by addressing thrombosis using the reported episodes of embolisms (respectively, pulmonary, arms, legs, central nervous system, heart, or other location) before, during, and 12 months after COVID-19 infection, according to health records and self-reported events in the survey. Neurologic evaluation was performed using a list of 25 pre-specified neurological symptoms [7]. Psychiatric evaluation was performed using four scores: the Center for Epidemiologic Studies Depression score [33,34], the State-Trait Anxiety Inventory score [35], the Pittsburgh Sleep Quality Index score [36], and the Maastricht Exhaustion Questionnaire [37]. Questionnaires and informed consent forms were sent each semester in 2021, and the last evaluation was obtained in December 2021. Survey responses contained no personally identifiable information, and addresses collected for survey distribution were encrypted as anonymized participant IDs. The survey consisted of 158 questions covering the different domains explored. The survey was created in French. The data included in the analysis were collected between March and December 2021.

### 2.3. Inclusion and Data Collection

All consecutive patients who were positive for SARS-CoV-2 infection after being tested at CHUV were assessed for recruitment in this observational study. CHUV’s electronic health records (EHR) provided epidemiological, clinical, radiological, and laboratory data. Epidemiological data included age, sex, height, weight, and relevant comorbidities (including Charlson Comorbidity Index (CCI)). We collected data on clinical presentation, onset of infection date, SARS-CoV-2 treatments, concomitant treatments, non-pharmacological interventions, and clinical course within CHUV. We calculated the quick Sequential Organ Failure Assessment (qSOFA) score, Confusion–Respiratory Rate–Blood Pressure–Age ≥ 65 Years (CRB-65) score, and National Early Warning Score (NEWS), according to their original descriptions [38,39,40].

The laboratory data included full blood count, D-dimers, creatinine, high-sensitivity cardiac troponin-T, C-reactive protein (CRP), procalcitonin (PCT), ferritin, and liver function tests.

We entered all data in an electronic clinical report form (eCRF) using the REDCap^®^ platform (Research Electronic Data Capture v8.5.24, Vanderbilt University, Nashville, TN, USA) [41]. Employees of the Infectious Diseases, Hospital Preventive Medicine, and Internal Medicine services at CHUV entered the data.

To compare the psychological characteristics of COVID-19 patients to those of the general population, we used the data of CoLaus|PsyCoLaus, a population-based prospective cohort study designed to assess the associations between mental disorders and cardiovascular risk factors among residents of the city of Lausanne [42,43]. The scores of the following self-rated scales could be compared between our patients and participants of CoLaus|PsyCoLaus: (1) The Center for Epidemiologic Studies Depression Scale (CES-D), a 20-item instrument developed for research in the general population to assess the severity of depressive symptoms over the past week on a 4-point scale [33,34]. (2) The state scale of the State–Trait Anxiety Inventory (STAI), a widely used and validated instrument for the evaluation of anxiety [35]. The test–retest reliability of the French version was 0.71–0.75 within 30 days and 0.65–0.68 within 60 days [44]. (3) The Pittsburgh Sleep Quality Index (PSQI), a questionnaire assessing sleep quality and disorders for the month preceding the evaluation [36]. Subjective sleep quality, sleep duration, efficiency, use of hypnotics, and poor daytime functioning are screened in this questionnaire. (4) The 9-item short version of the Maastricht Vital Exhaustion Questionnaire (MVEQ), which assesses vital exhaustion or burnout [37,45]. Items are related to undue fatigue, disturbed sleep, general malaise, irritability, loss of energy, and feelings of demoralization. The MVEQ was translated into French for the CoLaus|PsyCoLaus study.

### 2.4. Statistics

Statistical analyses were performed using R software v3.6.2 (R Foundation for Statistical Computing). Categorical variables are presented as numbers (percentages). Normally distributed continuous variables are presented as mean ± standard deviation (SD). Continuous variables with a skewed distribution are presented as median [interquartile range (IQR)].

For the descriptive analysis, we analyzed proportions of categorical variables using the Chi-squared goodness of fit test; we used the Student’s *t*-test for normally distributed variables; we used the Mann–Whitney–Wilcoxon test for continuous variables with a skewed distribution. We did not impute any values for missing data.

The analysis of the large number of neurological items collected 12 months after COVID-19 infection required data reduction using principal component analysis (PCA). Then, cluster analysis was applied to all the functional scores collected 12 months after infection, including the reduced neurological items, presuming that one of the clusters would gather the “Long COVID” cases. Once the clusters were identified, multivariable regression logistic models and stepwise backward selection according to the Akaïke criterion were employed to identify predictive patient characteristics both at baseline and during COVID infection.

PCA relies on the exploration of a correlation matrix and allows for data reduction with minimal loss of information [46]. It assembles variables into PCs if they share the same source of variability (i.e., are correlated), with the most significant variance being found in the first PC and subsequent orthogonal PCs (uncorrelated with the previous ones) encompassing less variance. The optimal number of PCs is identified by the scree test, which identifies the elbow on the plot of the amount of variability explained by each subsequent PC. Loadings are correlation coefficients between a PC and a variable and quantify the amount of information they share. To facilitate the interpretation, PCA employs varimax rotation, optimizing both the number of variables loaded on each PC and each variable’s load. Here, the solution was considered acceptable if it explained ≥50% of the total variability, if the PCs had ≥2 items with significant loadings, and cross-loading items (loadings ≥ 0.32 on 2 or more PCs) were dropped [47]. Items retained in the model had loadings ≥ 0.70 and significant contributions to both PCs (contribution of an item = variability explained by the item/variability explained by the PC × 100), i.e., above the expected average (if all items had uniform contributions) [48]. All functional scores and neurologic items collected at the time of COVID infection (quoted 1–7 or 1–4) were treated as quantitative, which made them most suitable for PCA.

Cluster analysis classifies the individuals into groups in order to maximize intra-group similarity and minimize inter-group similarity. Here, we used partitioning around medoids, a K-means algorithm that is more robust and less sensitive to outliers. Medoids are the most centrally located points in the cluster, i.e., individuals for which the average dissimilarity between them and all the other members of the cluster is minimal and who can be considered as representatives of the clusters. The silhouette (a measure of the average distance between clusters) and the gap statistic (a measure of intra-cluster variation) methods were used to identify the optimal number of clusters. The average silhouette width and the Dunn index (the ratio between the inter-cluster and the intra-cluster distances) were used for the validation of the cluster solution [49].

Missing values were imputed by the median of the observed values if they concerned adjustment variables in multivariable analyses and were neglected otherwise. The basic package of R software version 4.0.5 (https://cran.r-project.org/bin/windows/base/, accessed 13 April 2022) and the “factoMineR”, “factoextra”, “cluster”, and “fpc” libraries were used for all analyses. Statistical significance was set at *p* < 0.05.

### 2.5. Ethics

This project was conducted in accordance with the Declaration of Helsinki, the principles of Good Clinical Practice, and the Swiss Human Research Act (HRA). The project received approval from the Ethics Committee of Vaud canton, Switzerland (2020-01610). All data were anonymized before analysis.

## 3. Results

### 3.1. Demographics

A total of 1598 patients were selected and received the survey. Of these, 749 were ambulatory and 849 were hospitalized patients (Figure 1). Patients admitted in the hospital were significantly older than ambulatory patients (median 67 vs. 42 years old, respectively). The percentages of male and active or prior smokers were also significantly higher among hospitalized patients. An analysis of the comorbidities showed a higher percentage of the classical risk factors for COVID-19 severity in the hospitalized group, such as hypertension, obesity, and overweight; this was reflected by a difference in the Charlson Comorbidity Index (CCI) of hospitalized versus ambulatory patients (4 vs. 0, *p* < 0.001). These results are summarized in Appendix A.

We then compared the patients who answered the survey to the patients who did not answer the survey in the ambulatory cohort. We observed a lower percentage of responding men (34.9% vs. 42.5%) and fewer responses if the patient presented as overweight or obese. Other factors were not significantly different between the two groups (Appendix A). The same comparison was performed in the hospitalized group, of which 222 out of 849 patients (26%) answered our survey. The responding subgroup had a lower CCI than the non-responding subgroup (respectively 3.00 versus 4.00, *p* = 0.003). Diabetes, coronary disease, and chronic kidney disease were also significantly more frequent in the non-responding group (Appendix A). Our final cohort was therefore composed of 474 patients; 252 ambulatory and 222 hospitalized (Figure 1). Consistent with the initial analysis performed on the total population, the cohort analysis comparing ambulatory to hospitalized patients also found a significant difference in terms of age, gender, smoking habits, and comorbidities (Table 1).

### 3.2. Risks Factors Associated with Hospital Admission

Using multivariate analysis, we compared the ambulatory and hospitalized patients and analyzed the factors associated with hospital admission. Increasing age was associated with hospital admission, with the odds ratio reaching 3.3 for patients older than 50 years and peaking at 19 after 80 years of age. CCI score and smoking habits were also associated with increased risk of hospital admission (Table 2).

### 3.3. Cardio-Pulmonary Evaluation

Cardio-pulmonary evaluation was performed using the self-reported NYHA and CCS scales. As expected, the percentage of patients with an NYHA score measured at 1 (no limitation) was higher in the ambulatory group than in the hospitalized group (99.2% vs. 86.4%, *p* < 0.001) (Appendix A). A comparable result was observed when the CCS score was 0 (93.7% in ambulatory vs. 82.4% in hospitalized, *p* < 0.001). These scores were then reassessed at 1 year after COVID-19, and we evaluated the factors associated with a worsening of the NYHA score in multivariate analysis. Higher age was associated with worsening NYHA score (OR 1.01, 1.01–1.05, *p* = 0.015). Being female and having required ventilatory support were also significantly associated with a worse respiratory outcome at 1 year (OR at 2.31 and 3.72, respectively) (Appendix A). Regarding the CCS score, no significant factor was retrieved from the multivariate analysis.

### 3.4. Thrombotic Events

Thrombotic events were not frequently observed in our cohort; 94.5% of the patients during COVID-19 infection and 96.6% of patients at 1 year presented no thrombotic events. Thrombotic events were more frequent in the hospitalized cohort (Table 3). Pulmonary embolism (PE) during COVID was significantly associated with a higher risk of additive PE at 1 year; the percentage increased from 0.4% without COVID-PE to 11.1% if the patient had PE (*p* < 0.001). Similarly, any history of thrombosis before COVID was significantly associated with a higher proportion of thrombosis at 1 year (2.9% vs. 7.3%, *p* < 0.001). Focusing on different sites of thrombosis, the increased risk at 1 year compared to pre-COVID thrombosis occurrence was also statistically different for leg and cerebral thrombosis. All results are shown in Table 3.

### 3.5. Neurologic Evaluation

The most frequent symptom at COVID occurrence was fatigue, which was observed in 87.5% of patients, followed by “feeling slowed down”, headache, and smell disturbance in 71.5%, 68.5%, and 60.7% of cases, respectively. All the data are presented in Table 4 and Appendix A. Six variables improved significantly between COVID-19 occurrence and 1-year follow-up: headache, smell disturbance, taste disturbance, feeling slowed down, fatigue, and sleepiness (Appendix A). Analyzing the difference between during COVID-19 infection and 1 year post-infection, we observed that for each of these variables, except sleepiness, there was a greater improvement at 1 year for ambulatory patients compared to those who were hospitalized. The neurological parameters showing a significant difference in terms of improvement in hospitalized and ambulatory patients are illustrated in Figure 2. Moreover, age was also an important factor, as age > 65 was rarely associated with an improvement in any of these six factors (Appendix A). We performed multivariate analysis to determine the factors associated with the absence of improvement at 1 year. Higher age and comorbidities assessed by CCI score were associated with a higher risk of headache at 1 year; smell disturbance and taste disturbance were more likely to persist in patients with higher age or those who were hospitalized. Feeling slowed down and fatigue at 1 year were only more likely with increasing age (Table 5).

To include the neurologic symptoms in the clustering, we performed principal component analysis of the neurologic items suggested using the scree test to see whether a 3-dimensional solution would fit the data; however, the contributions to the 3rd PC were not sufficient, and the 2-dimensional solution was retained after the removal of four cross-loading variables. The items retained in the model are shown in bold in Appendix A. This will be included in the last part of our evaluation.

### 3.6. Psychological Evaluation

For the psychological evaluation, we compared our data to those of the population-based CoLaus|PsyCoLaus study [43]. In our cohort, among 18–49-year-old patients, only the scores of the CES-D, the STAI, and the PSQI could be compared with those of the participants of CoLaus|PsyCoLaus. Analyses were performed by sex and age with dichotomization at the age of 50 years.

Among 18–49-year-old men, neither the total sample nor the inpatient and outpatient subsamples differed from CoLaus|PsyCoLaus participants in terms of the total scores of the questionnaires (Appendix A). Among women in the same age range, our patients scored higher on the STAI than CoLaus|PsyCoLaus participants, regardless of whether they were inpatients or outpatients.

Regarding ≥50-year-old patients, differences with respect to people from the general population were more pronounced (Appendix A). Men in our cohort scored higher on all four questionnaires than the participants of CoLaus|PsyCoLaus. These differences were also observed for both the inpatient and outpatient subsamples of our cohort, except for the CES-D overall score and the PQSI in male outpatients. For women, we also observed higher scores for all four questionnaires in our cohort compared to the CoLaus|PsyCoLaus cohort. When splitting our cohort into inpatients and outpatients, differences remained for all scores in inpatients but only for the STAI and MVEQ scores in outpatients.

### 3.7. Clustering

The partitioning around medoids included all the functional scores, the CSD, MVEQ, STAI, and PSQI scores, as well as the selected items of the “Neurologic symptoms” and “Smell/Taste disturbance” dimensions (Appendix A) collected 12 months after COVID infection. Both the silhouette and gap statistics suggested that a two-cluster solution would fit the data. Clustering resulted in a “Long COVID” cluster, with *n* = 124, and an “Others” cluster, with *n* = 350. The average silhouette width was 0.41 and the Dunn index was 0.11. The discrimination between the clusters for the variables used for the partition was excellent, as shown in Appendix A. A visual representation of the clusters is shown in Figure 3.

The medoid of the “Long COVID” cluster was a 43-year-old woman who was a prior smoker with chronic obstructive lung disease and prior acute deep venous thrombosis; her CCI score was 1, her NYHA score was 1, her CCS score was 0, and she did not require hospitalization during COVID-19 infection. However, during COVID infection, she presented smell and taste disturbances rated as 7/7, as well as many neurological symptoms, such as concentration impairment with amnesia and fatigue (6/7). During the first year post-COVID, she presented another deep venous thrombosis episode and had altered functional scores at the 12-month assessment (CESD = 18, MVEQ = 10, STAI = 60).

The medoid of the “Others” cluster was a 62-year-old man with a body mass index of 23 who was a non-smoker, was treated with betablockers for previous dysrhythmia, and had experienced previous pulmonary fibrosis and polyneuropathy; his CCI score was 2, his NYHA score was 1, and his CCS score was 0. He did not report neurologic symptoms during COVID-19 infection but required hospitalization at day 4 and ended up on mechanical ventilation for acute respiratory distress syndrome. He developed deep venous thrombosis during COVID-19 infection as well as pulmonary thromboembolism with biological markers of severity (increased D-dimers, CRP). His functional scores at the 12-month assessment were lower than those of the other medoid (CESD = 8, MVEQ = 7, STAI = 25) and several of the neurological symptoms he reported at the 1-year evaluation indicated an increase in the severity of his prior polyneuropathy (balance disorder, hypoesthesia, paresthesia, fatigue).

Appendix A shows the baseline patient characteristics across clusters. Patients in the “Long COVID” cluster more often had prior chronic disease and had higher NYHA and CCS scores. No association was found between thromboembolism prior, during, or after COVID infection and membership of a certain cluster. Appendix A shows the patient characteristics during COVID-19 infection across clusters. Patients in the “Long COVID” cluster more often had neurologic symptoms and were more often admitted to hospital; they were also more often placed on mechanical ventilation. Table 6 shows the patient characteristics predictive of membership of the “Long COVID” cluster. The model had an adequate fit, with a R-squared coefficient of determination value of 0.51; had excellent discrimination, with a C-index of 0.89; and was well-calibrated, with a non-significant Hosmer–Lemeshow test (*p* = 0.68).

## 4. Discussion

This monocentric, longitudinal, observational, survey-based study assessed the persistence of multiorgan symptoms over a 12-month follow-up compared to the time of COVID-19 infection. In total, 1598 patients received the questionnaire, of which 474 responded (29.7%). Nearly half of our study cohort was hospitalized (222/474 patients). Unsurprisingly, the hospitalized population was significantly older, included more men, and had more active or past smoking habits, as well as a higher comorbidity index notably driven by overweight, obesity, and other features of the metabolic syndrome cluster. These data were consistent with the usual risk factors for hospitalization described in COVID-19 [50]. However, being male and overweight or obese were also significantly associated with not answering the questionnaire both for ambulatory and hospitalized patients, resulting in the significant underrepresentation of those risk factors in the final study population.

Focusing on cardio-pulmonary symptoms, hospitalized patients expectedly showed higher scores for dyspnea and angina during COVID-19 than ambulatory patients. At the 1-year follow-up, the risk factors associated with a higher NYHA score compared to baseline were age and patients having required mechanical ventilation. There was also a trend of women being at higher risk of having a worse NYHA score at 1 year without being statistically significant. Interestingly, these findings corroborated the observations of several authors concerning the risk factors of persistent dyspnea associated with lung diffusion impairment after COVID-19 [18,24].

It has been well documented that COVID-19 induces a hypercoagulable inflammatory state, leading to an increased incidence of thrombotic events and raising concerns about the need for an adequate thromboprophylaxis strategy in these patients [51,52]. Nevertheless, the expected duration of this hypercoagulable state, and thus the appropriate duration for primary or secondary prophylaxis, remains unclear and should be assessed with an individualized, balanced risk/benefit approach [53]. In this study, we found that patients with a history of thromboembolic event(s) prior to COVID-19 had a significantly higher risk of recurrence of thrombosis or embolism at 1 year. Moreover, patients diagnosed with PE during acute COVID-19 showed an incidence of additive PE of 11.1% during the year of follow-up. These numbers are probably driven by the overrepresentation of hospitalized patients in our cohort compared to the general population, but they highlight the importance of adequate management of anticoagulation after discharge.

We identified six neurological subjective complaints that were significantly different in prevalence between the time of acute COVID-19 diagnosis and 1-year follow-up. These symptoms were fatigue, feeling slowed down, headache, smell disturbance, taste disturbance, and sleepiness; of which the former four were also the most frequent symptoms experienced. All of these variables showed a trend towards improvement at the 1-year time point, but this improvement was significantly greater for ambulatory than for hospitalized patients. Factors associated with the persistence of neurological complaints at 1 year were age, comorbidities, and hospitalization. This was consistent with what has been previously described [15,16]. However, we did not find that females were significantly associated with a higher risk of neurologic sequelae, unlike some other authors [54]. For the other neurological complaints assessed, our results did not show any significant improvement at the 1-year time point. This is probably related to the lack of a control group for this parameter in our study; we were not able to determine whether they represented baseline, non-specific complaints present prior to COVID-19 or whether a proportion of them might represent Long COVID-associated, persistent neurologic sequelae. Interestingly, Xu et al. recently presented a cohort of 154,068 patients compared to contemporary and historical controls followed for neurologic disorders for 12 months following acute COVID-19 infection [55]. These authors showed an increased risk of a wide array of neurologic sequelae from episodic disorders such as migraines, which was consistent with our results, to more severe forms including ischemic and hemorrhagic stroke and encephalitis. SeeBle et al. also showed neurocognitive Long-COVID symptoms persisting after 1 year in a small cohort of 96 patients [56]. Fatigue was the most frequent symptom at 12 months, along with reduced exercise capacity and concentration problems. Another study performed in the Netherlands evaluated, in 76,422 patients, 23 somatic symptoms over time [57]. The Long COVID condition was defined as the persistence of core symptoms: chest pain, difficulties with breathing, pain when breathing, painful muscles, ageusia or anosmia, tingling extremities, lump in throat, alternately feeling hot and cold, heavy arms or legs, and general tiredness. The authors estimated that 12.7% of patients with COVID-19 would experience persistent symptoms. A large number of these symptoms were also found in our cohort.

To be able to use our data about neurologic symptoms in cluster analysis and allow for the visualization of these multidimensional features, we performed a principal component analysis that identified seven neurological items describing most of the variation in the dataset. The items were smell disturbance, taste disturbance, “pressure in the head”, feeling slowed down, “feeling as if in a fog”, concentration impairment, and memory impairment; smell and taste disturbance alone drove more than a half of the cumulative variability.

The incorporation of psychological scales into our study allowed us to compare the psychological scores of COVID-19 patients 1 year after infection with sex- and age-matched individuals of the general population (CoLaus|PsyCoLaus study). According to our data, the persistence of psychological symptoms (anxiety, fatigue, depression, and sleep disorder) was mostly limited to patients 50 years old and older, whereas in younger patients, mostly women were affected. These results confirmed that, as was previously described, there is considerable persistence of psychological symptoms over a 12-month period after COVID-19 infection in inpatients but also outpatients [58]. COVID-19 sequelae may persist longer than 12 months after the acute phase. However, these results need to be viewed with caution given that the psychological data of the comparison cohort stemmed from evaluations conducted prior to the COVID-19 pandemic. Therefore, these scores could not capture the profound multifactorial burden that the COVID-19 pandemic has represented on mental health in the global population since 2019 [59,60].

Based on our study, the final cluster analysis allowed us to identify two groups of patients, designated as the “Long COVID” and “Other” clusters. At the center of each cluster, we identified a medoid, represented by an individual whose characteristics showed the highest degree of similarity with the other members of his/her cluster, making him/her a “representative” patient.

Using this approach, we could also identify the clinical characteristics, at baseline or in the acute phase, that were predictive of belonging to the “Long COVID” cluster at 1 year. The baseline features that were significantly associated with “Long COVID” (Table 5) were the presence of chronic comorbidities—especially a history of atrial fibrillation or stroke—as well as the CCS score; being male was a protective factor. During acute infection, the neurological symptoms significantly associated with evolution into “Long COVID” were paresthesia, feeling slowed down, confusion, and insomnia. Patients who required mechanical ventilation were also more likely to belong to the “Long COVID” cluster. Strikingly, when analyzing the baseline profile of our “Long COVID” medoid in detail, we found that the patient had few of the pre-cited predictive factors (notably only neurological symptoms during COVID-19 infection). On the contrary, the medoid of the “Others” cluster had several risk factors (comorbidities, mechanical ventilation). These results highlight the fact that Long COVID is a complex heterogenous entity that cannot be overly simplified or predicted based to a stereotypical patient profile; it deserves to be assessed using an individualized approach.

Nevertheless, this study identified several factors that may play a role in and be associated with the persistence of long-term symptoms after acute COVID-19, such as age, female gender, hospitalization, mechanical ventilation, baseline comorbidities, and subjective neurological symptoms during the acute phase. These results are in agreement with a recent article published by Subramnian et al. [61]. In this very nice trial, the authors matched 486,149 adults infected with COVID-19 to 1,944,580 propensity score-matched adults. The multivariate analysis showed that female gender, belonging to an ethnic minority, socioeconomic deprivation, smoking, obesity and a wide range of comorbidities were risk factors for Long COVID. Some factors were not observed in our study because they were not included in the initial design (ethnic minority, socio economic deprivation).

We also draw attention to the long-term symptoms that can last for 12 months and probably more after acute illness, leading to an impact on patients’ lives and livelihoods. In an era when millions of individuals worldwide have recovered from COVID-19, and more will continue to be infected, there is still an unmet need for the improved characterization of the profile of Long COVID patients, to recognize these patients at an early stage, and, hopefully, to develop dedicated therapeutic and preventive approaches to allow for a faster recovery.

Our study had several limitations. Due to its design, being a single-center study performed in a tertiary hospital, it inevitably includes sampling bias, illustrated notably by the overrepresentation of hospitalized patients. Furthermore, by screening only patients with confirmed SARS-CoV-2 infection and who were discharged alive, the study poses a risk of both under-coverage bias and survivorship bias. Because we used a survey-based approach with questionnaires created only in French sent in the mail, we also created a voluntary response bias in addition to excluding patients unable to respond due to either a language barrier (possibly leading to an additional ethnicity-based or sociocultural bias) or neurocognitive disorders. These limitations are reflected in the low participation rate. Moreover, the need for a retrospective assessment of the symptoms present at the time of COVID-19 infection creates a possible recall bias, which could lead to both underreporting and overreporting of symptoms. Finally, due to the timing of the study recruitment, we could not assess the potential impact of variants of concern and vaccination on Long COVID. Moreover, treatment changed a lot between the first wave and subsequent episodes with the key roles of corticosteroids as well as anticoagulants [62,63], both of which may influence the development of long COVID. For all of these reasons, our cohort might have limited representativeness when it comes to the average Long COVID patient and possibly failed to fully capture the wide spectrum of the severity of the disease.

## 5. Conclusions

Evaluating and reporting subjective parameters such as fatigue or weakness remains challenging, and there is a crucial need for standardized research tools to be able to gather robust evidence.

## Figures and Tables

**Figure 1 jcm-12-02673-f001:**
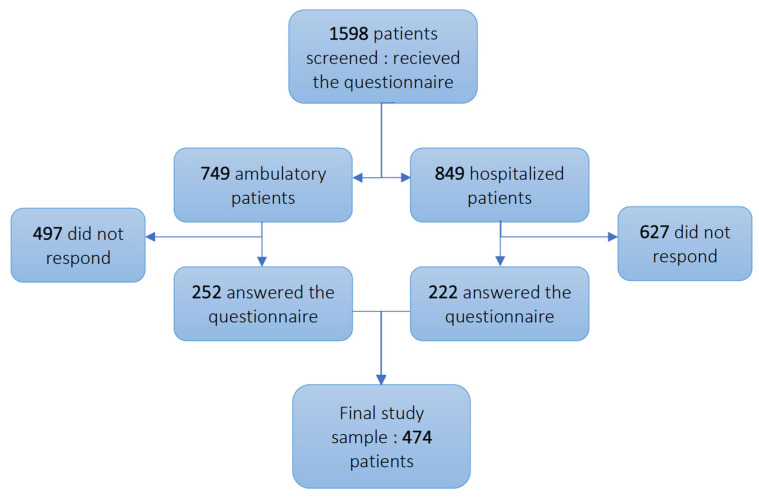
Flowchart of the study population.

**Figure 2 jcm-12-02673-f002:**
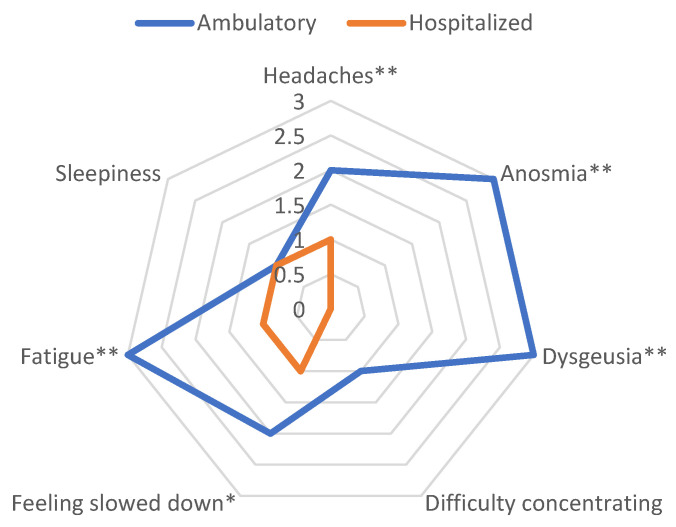
Delta score measuring the improvement of neurological variables between COVID occurrence and 1-year follow-up. Each variable is treated as a difference of median. Values are compared using Wilcoxon rank sum test. * *p* < 0.05, ** *p* < 0.001.

**Figure 3 jcm-12-02673-f003:**
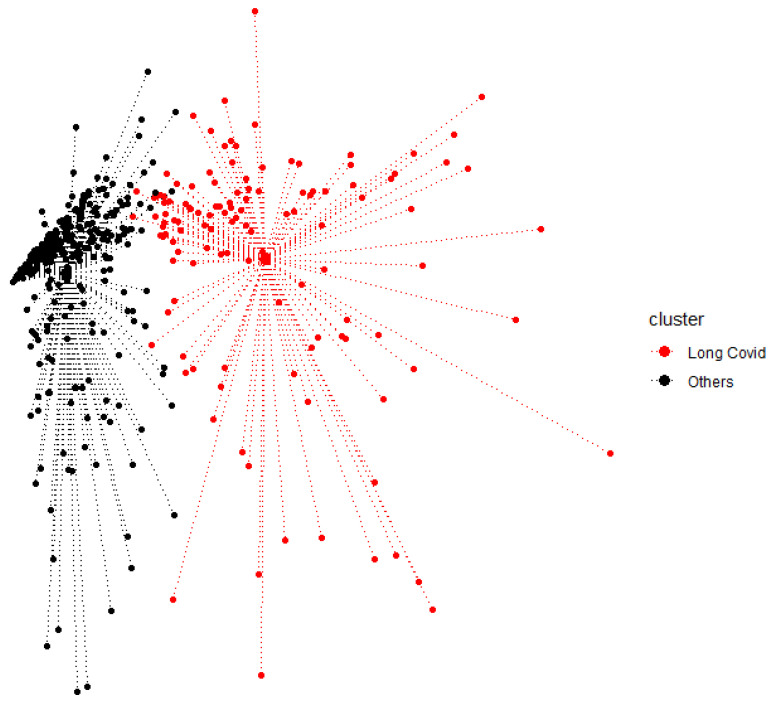
Two-dimensional representation of the clusters. The starplot centers represent the medoids.

**Table 1 jcm-12-02673-t001:** Characteristics of the study cohort of responding COVID-19 patients.

	Ambulatory, N = 252 ^1^	Hospitalized, N = 222 ^1^	Overall, N = 474 ^1^	*p*-Value ^2^
Age	41 (31, 56)	66 (55, 78)	55 (38, 69)	<0.001
Gender				<0.001
Men	34.9% (88)	55.4% (123)	44.5% (211)	
Women	65.1% (164)	44.6% (99)	55.5% (263)	
Cigarette smoking				<0.001
Never smoker	93.5% (229)	80.2% (178)	87.2% (407)	
Former smoker	4.1% (10)	17.1% (38)	10.3% (48)	
Current smoker	2.4% (6)	2.7% (6)	2.6% (12)	
Unknown	7	0	7	
Comorbidities				
Charlson Comorbidity Index	0.00 (0.00, 2.00)	3.00 (2.00, 5.00)	1.50 (0.00, 4.00)	<0.001
Diabetes	4.0% (10)	19.4% (43)	11.2% (53)	<0.001
Cirrhosis	0.0% (0)	0.9% (2)	0.4% (2)	0.2
Cancer	3.6% (9)	6.3% (14)	4.9% (23)	0.2
Obesity	2.8% (7)	30.6% (68)	15.8% (75)	<0.001
Overweight and Obese	7.1% (18)	60.8% (135)	32.3% (153)	<0.001
Hypertension	13.9% (35)	45.9% (102)	28.9% (137)	<0.001
Atrial Fibrillation	0.4% (1)	9.5% (21)	4.6% (22)	<0.001
Coronary Disease	2.8% (7)	9.0% (20)	5.7% (27)	0.003
Stroke	1.6% (4)	6.3% (14)	3.8% (18)	0.007
Chronic Kidney Disease	0.4% (1)	8.1% (18)	4.0% (19)	<0.001
COPD	0.4% (1)	7.7% (17)	3.8% (18)	<0.001
Asthma	9.1% (23)	5.4% (12)	7.4% (35)	0.12
Ventilation status				<0.001
Not Ventilated	100.0% (252)	87.4% (194)	94.1% (446)	
Ventilated	0.0% (0)	12.6% (28)	5.9% (28)	

^1^ Median (IQR); % (N) ^2^ Wilcoxon rank sum test; Pearson’s Chi-squared test; Fisher’s exact test.

**Table 2 jcm-12-02673-t002:** Risks factors associated with hospital admission.

	OR ^1^	95% CI ^2^	*p*-Value
Age group			
18–49	1.00		
50–64	3.30	1.75, 6.25	<0.001
65–79	3.19	1.24, 8.11	0.015
>80	19.0	3.65, 149	0.001
Charlson comorbidity index	1.39	1.15, 1.72	0.001
Cigarette smoking			
Never smoker	1.00		
Former smoker	2.69	1.17, 6.60	0.024
Current smoker	1.94	0.46, 7.79	0.4

^1^ OR: odds ratio, ^2^ CI: confidence interval.

**Table 3 jcm-12-02673-t003:** Occurrence of thrombotic events.

Thrombotic Events	Overall Population (N = 474) ^1^		
Location	History at Baseline	COVID Occurrence	1-Year Follow-Up
No embolism	91.4% (433)	94.5% (448)	96.6% (458)
Pulmonary	2.1% (10)	3.8% (18)	0.8% (4)
Upper limbs	0.4% (2)	0	0.4% (2)
Central nervous system	1.5% (7)	0.4% (2)	0
Coronary	1.7% (8)	0.8% (4)	0.6% (3)
Lower limbs	4.6% (22)	1.5% (7)	1.5% (7)
Embolism or thrombosis of other location	0.8% (4)	0.2% (1)	0.4% (2)
Thrombotic events	Hospitalized patients (N = 222) ^1^		
Location	History at baseline	COVID occurrence	1-year follow-up
No embolism	87.4% (194)	90.1% (200)	93.7% (208)
Pulmonary	3.6% (8)	7.2% (16)	1.8% (4)
Upper limbs	0.5% (1)	0	0.9% (2)
Central nervous system	2.3% (5)	0.9% (2)	0
Coronary	1.8% (4)	1.8% (4)	1.4% (3)
Lower limbs	6.8% (15)	2.3% (5)	2.7% (6)
Embolism or thrombosis of other location	1.4% (3)	0.5% (1)	0.5% (1)
Thrombotic events	Ambulatory patients (N = 252) ^1^		
Location	History at baseline	COVID occurrence	1-year follow-up
No embolism	94.8% (239)	98.4% (248)	99.2% (250)
Pulmonary	0.8% (2)	0.8% (2)	0
Upper limbs	0.4% (1)	0	0
Central nervous system	0.8% (2)	0	0
Coronary	1.6% (4)	0	0
Lower limbs	2.8% (7)	0.8% (2)	0.4% (1)
Embolism or thrombosis of other location	0.4% (1)	0	0.4% (1)

^1^ % (N).

**Table 4 jcm-12-02673-t004:** Neurological symptoms at COVID occurrence.

	Ambulatory, N = 252 ^1^	Hospitalized, N = 222 ^1^	Overall, N = 474 ^1^	*p*-Value ^2^
Fatigue	90.8% (229)	83.7% (186)	87.5% (415)	0.3
Anosmia	75.8% (191)	43.6% (97)	60.7% (288)	<0.001
Dysgeusia	70.2% (177)	41.8% (93)	56.9% (270)	<0.001
Headaches	76.9% (194)	59.0% (131)	68.5% (325)	<0.001
Feeling slowed down	71.0% (179)	72.0% (160)	71.5% (339)	0.4
Sleepiness	56.7% (143)	60.3% (134)	58.4% (277)	>0.9
“Pressure in the head”	57.1% (144)	41.4% (92)	49.7% (236)	0.032
“Feeling as if in a fog”	46.0% (116)	54.9% (122)	50.2% (238)	0.2
Difficulty focusing	61.1% (154)	59.4% (132)	60.3% (286)	>0.9
Confusion	22.2% (56)	39.6% (88)	30.3% (144)	0.001
Memory problems	32.5% (82)	53.6% (119)	42.4% (201)	<0.001
Insomnia	27.3% (69)	47.7% (106)	36.9% (175)	<0.001

^1^ % (N) ^2^ Pearson’s Chi-squared test.

**Table 5 jcm-12-02673-t005:** Factors associated with the absence of improvement at 1 year by multivariate analysis.

	OR ^1^	95% CI ^2^	*p*-Value
Headaches			
● Age	1.02	1, 1.04	0.015
● Sex			
○ Men	1.00		
○ Women	0.88	0.58, 1.35	0.6
● Charlson Comorbidity Index	1.21	1.05, 1.40	0.012
Anosmia			
● Age	1.02	1.01, 1.04	<0.001
● Status			
○ Ambulatory	1.00		
○ Hospitalized	2.77	1.75, 4.34	<0.001
Dysgeusia			
● Age	1.02	1.01, 1.03	<0.001
● Status			
○ Ambulatory	1.00		
○ Hospitalized	2.27	1.45, 3.57	<0.001
Feeling slowed down			
● Age	1.01	1, 1.02	0.005
Fatigue			
● Age	1.02	1.01, 1.03	<0.001

^1^ OR: odds ratio, ^2^ CI: confidence interval.

**Table 6 jcm-12-02673-t006:** Predictive model for Long COVID. All variables significantly associated with “Long COVID” cluster membership in the univariable analysis (*p* < 0.10) were included in the multivariable prediction model. Stepwise regression was performed with backward selection of the variables retained in the model according to the Akaike criterion.

Predictive Variable		Odds Ratio	95% Confidence Interval	*p*-Value
Male		0.89	(0.84–0.95)	<0.001
Baseline:				
	CCS score	1.14	(1.06–1.23)	<0.001
	Prior chronic disease	1.07	(1.006–1.15)	0.03
	Stroke	1.23	(1.04–1.45)	0.01
	Atrial fibrillation	1.19	(1.03–1.39)	0.02
During COVID infection				
	Tingling, burning sensation	1.06	(1.03–1.08)	<0.001
	Feeling slowed down	1.02	(1.003–1.04)	0.02
	Memory problems	1.03	(1.01–1.05)	0.001
	Confusion	1.04	(1.02–1.07)	<0.001
	Insomnia	1.02	(1.007–1.04)	0.006
	Mechanical ventilation	1.23	(1.07–1.41)	0.003

## Data Availability

Data are available on demand to the corresponding author.

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
