# Peer review of "Long-Term Consequences of COVID-19: A 1-Year Analysis"

_jcm, 2023, doi:10.3390/jcm12072673_

Round 1

Reviewer 1 Report

This is an important and large study attempting to identify and evaluate long-covid symptoms. The authors recruited covid-19 patients during the first wave in 2020 and performed a follow-up evaluation 1-year later. Main organ systems affected by covid-19 were evaluated both in first visit and in follow-up including cardiopulmonary, nervous system. Psychological evaluation using standardized questionnaires was also done.

Results from data analysis demonstrate patient characteristics linked to hospitalization. Main cardiopulmonary and neurological symptoms are clearly described. Cluster analysis depicts the profile of the so-called long-covid patient, which largely has been in question so far. Thrombotic predisposition of covid-19 patients was also evaluated and found similar to previous studies.

A limitation of the study that is already stated by authors themselves is that the circulating virus at the time of hospitalization was the wild strain, therefore SARS-CoV-2 variant symptoms could not be studied. Moreover, the impact of vaccination on symptoms could not be identified either in this study.

In conclusion, this is a well-designed study including a large population leading to interesting findings.

Author Response

This is an important and large study attempting to identify and evaluate long-covid symptoms. The authors recruited covid-19 patients during the first wave in 2020 and performed a follow-up evaluation 1-year later. Main organ systems affected by covid-19 were evaluated both in first visit and in follow-up including cardiopulmonary, nervous system. Psychological evaluation using standardized questionnaires was also done.

Results from data analysis demonstrate patient characteristics linked to hospitalization. Main cardiopulmonary and neurological symptoms are clearly described. Cluster analysis depicts the profile of the so-called long-covid patient, which largely has been in question so far. Thrombotic predisposition of covid-19 patients was also evaluated and found similar to previous studies.

A limitation of the study that is already stated by authors themselves is that the circulating virus at the time of hospitalization was the wild strain, therefore SARS-CoV-2 variant symptoms could not be studied. Moreover, the impact of vaccination on symptoms could not be identified either in this study.

In conclusion, this is a well-designed study including a large population leading to interesting findings.

We thank this reviewer for these nice remarks about our study.

Reviewer 2 Report

The manuscript entitled “Long Term Consequences of COVID-19, A 1-Year Analysis” is well written and well presented. I have few minor concerns.

Results. Demographics. Line Line 244 -262.

Demographics of patients are well defined under the heading, from line 244-262. Same data is shown in flowchart as figure 1. It is data repetition. I will suggest to decrease the explanation about the number of patients in different groups, who responded or not in the lines 244-262 and keep the flow chart in figure 1, which is more valuable.

Table 1, row 3. Please replace the word Sex with Gender.

The major comorbidities are Diabetes and hypertension. Do you find any relationship of these two comorbidities with hospital admission or long covid?

Heading Thrombotic events. During COVID-19, different treatment options were used for COVID-19 patients. Do the patients treated at your hospital were given anticoagulants? or some of them were already on anticoagulants? There are reports of good response of anticoagulants and steroids during covid infection. You may add a paragraph in introduction and explain the type of drugs which were given to covid patients during the covid infection.

https://pubmed.ncbi.nlm.nih.gov/34827332/

https://pubmed.ncbi.nlm.nih.gov/34943722/

There is no need for sub headings under the discussion heading.

There are many paragraphs in the discussion which are not properly discussed with the already published data. Add more references in discussion and discuss your results properly.

Author Response

The manuscript entitled “Long Term Consequences of COVID-19, A 1-Year Analysis” is well written and well presented. I have few minor concerns.

Results. Demographics. Line Line 244 -262.

Demographics of patients are well defined under the heading, from line 244-262. Same data is shown in flowchart as figure 1. It is data repetition. I will suggest to decrease the explanation about the number of patients in different groups, who responded or not in the

lines 244-262 and keep the flow chart in figure 1, which is more valuable.

We changed this section as suggested by the reviewer and removed the numbers which are in figure 1 as underlined

Table 1, row 3. Please replace the word Sex with Gender.

We changed the word as suggested in the text and the table

The major comorbidities are Diabetes and hypertension. Do you find any relationship of these two comorbidities with hospital admission or long covid?

These 2 factors were included in the statistical analysis for both hospital admission and long COVID, none of them was statistically associated with one or the other.

Heading Thrombotic events. During COVID-19, different treatment options were used for COVID-19 patients. Do the patients treated at your hospital were given anticoagulants? Or some of them were already on anticoagulants? There are reports of good response of anticoagulants and steroids during covid infection. You may add a paragraph in introduction and explain the type of drugs which were given to covid patients during the covid infection.

Our study was performed during the first wave of COVID, at this time there was no clear recommendation in our institution regarding anticoagulants nor steroids which were implemented in a second phase after the results of platform trials. For this reason, this parameter was not clearly evaluated per se and we did not discuss it. We nevertheless agree with this reviewer on the key role of both steroids for severe, and anticoagulants for COVID 19 patients. This was added in the limitations.

There is no need for sub headings under the discussion heading.

We removed the headings in the discussion section

There are many paragraphs in the discussion which are not properly discussed with the already published data. Add more references in discussion and discuss your results properly.

This remark is correct and we added more recent references published on the topic in the discussion section as suggested.
